# The Effect of Terminal Freezing and Thawing on the Quality of Frozen Dough: From the View of Water, Starch, and Protein Properties

**DOI:** 10.3390/foods12213888

**Published:** 2023-10-24

**Authors:** Xiaorong Liu, Luncai Chen, Lei Chen, Dezheng Liu, Hongyan Liu, Dengyue Jiang, Yang Fu, Xuedong Wang

**Affiliations:** 1Key Laboratory for Deep Processing of Major Grain and Oil, Ministry of Education, Wuhan Polytechnic University, Wuhan 430023, China; 18253562371@163.com (X.L.); chenleiy@whpu.edu.cn (L.C.); 15337153248@163.com (H.L.); 2Hubei Key Laboratory for Processing and Transformation of Agricultural Products, Wuhan Polytechnic University, Wuhan 430023, China; chenluncai2000@163.com (L.C.); jiangdy1217@163.com (D.J.); 3Hubei Selenium Grain Technology Group Co., Ltd., Enshi 445600, China; zzh2008214@163.com

**Keywords:** frozen dough, freeze–thaw, texture, water state, β-sheet

## Abstract

Frozen dough is suitable for industrial cold chain transportation, but usually experiences temperature fluctuations through the cold chain to the store after being refrigerated in a factory, seriously damaging the product yield. In order to analyze the influence mechanism of temperature fluctuation during the terminal cold chain on frozen dough, the effects of terminal freezing and thawing (TFT) on the quality (texture and rheology) and component (water, starch, protein) behaviors of dough were investigated. Results showed that the TFT treatment significantly increased the hardness and decreased the springiness of dough and that the storage modules were also reduced. Furthermore, TFT increased the content of freezable water and reduced the bound water with increased migration. Additionally, the peak viscosity and breakdown value after TFT with the increased number of cycles were also increased. Moreover, the protein characteristics showed that the low-molecular-weight region and the β-sheet in the gluten secondary structure after the TFT treatment were increased, which was confirmed by the increased number of free sulfhydryl groups. Microstructure results showed that pores and loose connection were observed during the TFT treatment. In conclusion, the theoretical support was provided for understanding and eliminating the influence of the terminal nodes in a cold chain.

## 1. Introduction

In the last decade, the food industry has an increasing requirement for frozen dough, which is widely used to produce bread, cereal food, etc. Frozen dough has important implications for the industrialization of modern wheat-based staples, with significant benefits for both factories and consumers [1]. On the one hand, frozen dough facilitates food processing, storage, and selling, which can significantly increase the shelf life. On the other hand, frozen dough technology allows consumers to enjoy fresh bread [2]. Frozen dough can be mass-produced in a central factory and then transported through a cold chain to local bakeries and stores for on-site baking, which can save the costs of equipment and labor. However, frozen storage in a cold chain also destroys the dough structure, affecting the final quality of the dough-making food, such as increased hardness and reduced specific volume of bread [3].

The core issue in the cold chain transportation of dough-based food is temperature fluctuation, which has a direct impact on the final quality of food [4]. In response to the change in the temperature and time of frozen storage, the uneven internal structure of the product causes cracking and shrinkage, which severely restrict the large-scale application of frozen dough. During frozen storage, the quality of frozen dough gradually decreases, which is closely linked to the formation and changes in the dough network [5]. There are various components that make up dough, including water, gluten, and starch. The production process of dough involves mixing wheat flour with water to form gluten, which interact with each other to form a three-dimensional network [6]. Through this structure, the dough would retain water and have a viscoelastic structure. During storage with temperature fluctuation, dough-based foods may suffer serious quality problems due to ice crystal growth and recrystallization. It was found that frozen dough loses water after freezing and thawing and that large ice crystals in the dough destroy the internal structure, leading to the separation of starch granules and the protein network structure [7]. Moreover, the water distribution and microstructure of frozen dough can be altered by multiple freeze–thaw cycles, for example, promoting ice recrystallization, disrupting the interaction between dough components and water molecules, and depolymerizing the high-molecular-weight portion of gluten [8].

Nowadays, the temperature maintenance technology of most cold chain trucks or cold chain warehouses in the main cold chain transportation system is developing rapidly and temperature control is maintained better and better [9]. On the contrary, during terminal cold chain transportation, such as during the handling and placement process, temperature changes are more drastic. But, as far as we know, the impact of freezing and thawing during the terminal cold chain on the dough quality is very often overlooked [10]. Temperature fluctuation may occur during transportation at the end of the cold chain. This is called scission in terminal chain. In the cold chain transportation process, terminal chain scission seriously affects the quality of food, resulting in a lower yield of dough product. There are several studies that focused on freezing and freeze–thawing during cold chain storage [11,12]. Nevertheless, less systematical research exists on the differences in quality changes after this composite treatment (freeze–thaw after frozen storage) rather than individual freezing or freeze–thaw.

Overall, it was hypothesized that the differences in the key components of frozen dough could further explain the reason for low yield under TFT compared to fresh and single-frozen storage. In light of this, we investigated the effects of terminal freezing and thawing on the texture and dynamic rheology in frozen dough. Then, we analyzed the water properties (freezable water, moisture distribution) and starch pasting properties of frozen doughs. Especially the protein properties (molecular weight and protein secondary structure) were analyzed to further investigate the polymerization of gluten in the TFT-treated doughs. The study outcome helps us to understand the changes in the physicochemical quality of frozen dough during terminal chain scission and will provide guidance to an industry interested in frozen dough during cold chain transportation.

## 2. Materials and Methods

### 2.1. Materials

Golden Statue Brand wheat flour (LamSoon Limited, Hong Kong, China) and high-activity dry yeast (Angel Yeast Co., Ltd., Yichang, China) were bought from the supermarket. Analytical-grade reagents were used for all other experiments.

### 2.2. Preparation and TFT Treatment of Dough

The production of dough in this study was based on our previous study [1]. In the formulation, there were the following components: wheat flour, yeast (1.5%, based on the wheat flour), sodium chloride (1%), and water (about 60%). After mixing for 12 min, the ingredients rested at room temperature (25 °C) for 10 min. After being divided into 80 g, the dough was placed in a rapid freezer (JB1-5F, Shanghai Jincheng Refrigeration Equipment Co. Ltd., Shanghai, China) at −30 °C for 30 min, named as FR dough (fresh dough). Then, the FR dough was frozen at −20 °C (DW-40 L188, Medical cryopreservation box, Qingdao, China) for 30 d, named as FS (frozen storage, which simulated refrigerated transportation). Then, the FS dough was frozen at −20 °C for 20 h and then thawed at 25 °C for 1 h, considered as one cycle, named as TFT-1 (terminal freezing and thawing). The FS dough treated with 2, 4, and 6 cycles of freeze–thaw was named as TFT-2, TFT-4, and TFT-6, which were used in this study to simulate the scission in a terminal cold chain.

### 2.3. Texture Properties

The hardness and springiness of dough were measured using a texture analyzer (TA.XTC-20, Bosin, Shanghai, China) with a TA/LKB probe. The parameters were as follows: the test speed was 1 mm/s and compression ratio was 50%, repeated three times and used for each group.

### 2.4. Dynamic Rheological Properties

A DHR-2 rheometer (TA Instruments, DE, New Castle, USA) was used to detect the dynamic rheological properties. The angular frequency of dynamic scanning was 1–100 rad s^−1^, and the parameter of strain amplitude was 1%. The storage modulus (G′) and loss modulus (G″) were measured [13].

### 2.5. The Content of Freezable Water 

The amount of freezable water in the dough was determined by DSC (differential scanning calorimetry, Q2000, TA Instruments, USA). The fresh dough powder (10–15 mg) was weighed, sealed, and balanced at −30 °C for 5 min and then heated from −30 °C to 30 °C at 10 °C/min. An empty crucible was used as the reference. The melting enthalpy (ΔH) was recorded, and the freezable water content (FW) in the frozen dough could be calculated by Equation (1) [14].
FW (%) = ΔH/(ΔHi × WA) × 100%(1)FW was the content of freezable water in each sample, ΔH was the melting enthalpy (J/g), ΔHi was the latent heat from the fusion of ice, which was 334 J/g, and WA was the water content (g/g).

### 2.6. Water Distribution

The water distribution was determined by LF-NMR (low-field nuclear magnetic resonance, Niumag, Shanghai, China). For the LF-NMR test, in the process of acquiring the transverse relaxation curves, Carr–Purcell–Meiboom–Gill sequences were used [15]. The relaxation time and relative peak area were recorded.

### 2.7. Pasting Properties

A rapid viscosity analyzer (RVA-Super 4, Perten Instruments, Hudinge, Sweden) was used to determine the pasting properties. The dough samples were heated at 50 °C (1 min), raised to 95 °C with 12 °C/min, and maintained at 95 °C (2.5 min); then, it was cooled back down to 50 °C and maintained for 2 min. The pasting parameters were obtained by RVA software (TCW 3.14).

### 2.8. Protein Parameters

#### 2.8.1. SDS-PAGE

SDS-PAGE was slightly modified according to our previous method [16]. The doughs with different treatments were washed in 20 g/L NaCl and distilled water to obtain gluten. Then, the gluten was freeze-dried and was dissolved in Tris-HCl buffer (0.125 mol/L, pH 6.8) containing SDS (2%, *w*/*v*), glycerol (10%, *v*/*v*), and bromophenol blue (0.01%, *w*/*v*). The protein was extracted at room temperature and centrifuged (10,000× *g*, 20 min), from which the supernatant was obtained. The supernatant was diluted with 5 × loading buffer and boiled in the water bath for 5 min. The 5% stacking gel (pH 6.7) and 12% resolving gel (pH 8.8) were used as part of the gel system. Supernatants were loaded and concentrated at 80 V and separated at 120 V.

#### 2.8.2. Free Thiol Group (SH) Content

The detection of the free thiol group content was based on a previous report [6]. The freeze-dried protein sample was added into Tris-glycine buffer (8.0 M urea, 1.0 mM EDTA, pH 8.0) and mixed well; this was shaken (1 h) and centrifugated (6000× *g*, 15 min). Then, 5,5′-dithiobis (2-nitrobenzoic acid) was incorporated into the supernatant and shaken. The absorbance was measured at 412 nm by spectrophotometer (UV-3600, Shimadzu Co., Kyoto, Japan) and divided by the molar absorption (13,600) to obtain the free SH content.

#### 2.8.3. Protein Secondary Structure

Fourier transform infrared spectroscopy (FTIR, NEXUS670, Waltham, MA, USA) was used to determine the protein secondary structure. The wavenumber ranged from 4000 to 400 cm^−1^. Each measurement was scanned 32 times and the resolution was 4 cm^−1^. OMNIC v8.2 and Peakfit software v4.12 were used to measure the data [17].

### 2.9. SEM

The dough samples were freeze-dried in a vacuum freeze-dryer. Then, the dough was gold-sprayed and the sample was observed by SEM (scanning electron microscope, Hitachi Co., Ltd., Beijing, China) with a 1000× magnification.

### 2.10. Statistical Analysis

Statistical analysis was performed with SPSS19.0 analysis software using one-way analysis of variance (ANOVA) and Duncan’s multiple range test (*p* < 0.05).

## 3. Results and Discussion

### 3.1. The Texture Properties of the Dough after TFT Treatment

Hardness and springiness are typical indicators of texture, which can initially measure the quality of dough [18]. We investigated the effect of terminal chain scission on the dough quality and the hardness and springiness of the dough under frozen storage (FS) and terminal freezing and thawing (TFT) treatments; this is summarized in Figure 1A,B.

The effect of FS and TFT on the texture of the dough compared to that of the control (FS) was significant (*p* < 0.05). The hardness of the dough after 1 month of frozen storage was more than that of the FR dough, and the increase was greater with the increased number of terminal freezing and thawing cycles. As shown in Figure 1A,B, the hardness drastically increased after the TFT treatment. The lesser hardness reflected the softer dough at a certain range. The increase in hardness indicated that TFT significantly affected the dough compared to the FS treatment. On the one hand, with the increased number of cycles of TFT, the water loss accelerated. On the other hand, the recrystallization of ice crystals during TFT caused mechanical damage to the dough, destroying the dough contents, thus affecting the texture of the dough [19]. Meanwhile, the dough springiness after the TFT treatment was significantly less than that of FR and FS, indicating that the TFT treatment reduced the dough extension and accelerated the deterioration of texture [20].

In our study, the hardness and springiness change after the TFT treatment was more significant than that after FS, indicating that the negative effect of TFT on dough quality was more serious than FS. To further explain the deterioration mechanism of TFT on dough quality, more analysis was necessary.

### 3.2. Dynamic Rheology Property

Viscoelasticity was the main factor that determined the final quality of the dough, and the storage modulus (G′) and loss modulus (G″) in the dynamic rheology test could describe the elastic and viscous status of the dough [21].

To better explain how the TFT treatment affected the overall mechanical properties, Figure 1C,D shows the dynamic rheological data of the frozen dough after FS and TFT treatments. Based on the results of the frequency range, the storage module (G′) and loss module (G″) showed an increase trend, and the increase in G′ was much more than that of G″. This result suggested that the dough was elastic rather than sticky [22]. With the TFT treatment, the G′ and G″ showed a decreased trend compared to the FS sample, indicating that terminal freezing and thawing destroyed the dough viscoelasticity. This result was also supported by our texture data of an increase in hardness and decrease in springiness of the dough. Meanwhile, a comparison was investigated between FS and TFT. Compared with FS, the reduction in TFT was significantly higher and only two cycles of freezing and thawing exceeded the difference of FS. The results showed that the drastic change in rheological properties caused by TFT was greater than that caused by FS and that the TFT treatment severely weakened the structure of the dough.

The difference in dynamic rheology was mainly determined by the internal components of the dough. The spatial structure of the dough network was mainly composed of gluten proteins in which starch and water were filled in [23]. With the treatment of TFT, the compaction degree of the frozen dough was higher, which may have been the result of the water recrystallization in the dough system. The regrowth of ice crystals caused mechanical damage to the dough network, leading to the reduction in starch–gluten crosslinking [8]. Additionally, the starch properties and protein changes caused by the ice during the TFT treatment may directly affect the dough rheological properties; further analysis is needed.

### 3.3. Freezable Water

Water is an indispensable component of frozen dough and plays an essential role in it. The freezable water reflected the amount of ice in the frozen dough and was one of the major contributing factors affecting the quality of the frozen dough [24]. When freezable water was frozen into ice crystals, the frozen dough was mechanically damaged by the growth and formation of ice crystals. Therefore, in this study, DSC was used to analyze the change in ice crystal melting enthalpy (ΔH) and freezable water of the dough.

As shown in Figure 2A,B, based on our results, we found that ΔH and freezable water of the frozen dough increased under FS and TFT treatments. This was because the dough network was destroyed when exposed to the FS and TFT treatments. As ice crystals recrystallized during frozen storage, amylose and amylopectin changed in structure and arranged, resulting in an increase in ΔH [25]. Meanwhile, the FS treatment increased the freezable water content in the frozen dough and significantly increased with the number of cycles of TFT. The freezable water content after the FS treatment was 52.97% (increased by 3.70% relative to the FR). The freezable water content in TFT increased to 62.9% after only two freeze–thaw cycles (increased by 9.93% compared to FS), and this percentage increased with the number of cycles. From two to six cycles of TFT, ΔH and freezable water increased continuously, showing that the effect of TFT on dough disruption was much higher than that of the FS treatment. This may have been due to the continuous recrystallization of ice during the TFT, leading to the separation of water molecules in the dough and the destruction of the gluten network [26]. The binding of the water molecule to the protein in the dough may have been broken, resulting in the release of part of the bound water.

These results showed that TFT would sharply increase the content of freezable water compared to the FS treatment with a constant temperature in the frozen dough. This indicated that the TFT treatment disrupted the stability of the dough system, which was induced by increasing the number and volume of ice crystals through terminal freeze–thaw cycles in the TFT treatment, intensifying the physical damage to the gluten network.

### 3.4. The Water Migration of the Dough after TFT Treatment

The stable gluten network in the dough was formed with water by interacting with various components in flour, so it was essential to study the water status and distribution to understand the properties of the dough [27]. A recent study showed that it was the dynamic migration of water molecules in food that directly affected the stability of food [11]. In order to further investigate the water migration in the dough, LF-NMR was used to measure the change in water status under the TFT treatment.

Three relaxation times (T21, T22, T23) were detected in the frozen dough with FS and TFT treatments, which represented bound water, immobilized water, and free water, respectively. A21, A22, and A23 indicated the three kinds of water proportions in frozen dough, respectively. As shown in Figure 2C, after the FS treatment, A21 decreased from 20.31% to 19.71%, while A22 (from 75.95% to 76.14%) and A23 (from 3.74% to 4.41%) increased. In addition, the A21 under two cycles of the TFT treatment decreased from 19.71% to 18.23%. Thus, we speculated that FS and TFT treatments could reduce the bound water and promote the migration of water toward the more mobile free water. Among them, the changes in tightly bound water were mainly caused by the interaction between protein and water molecules in the frozen dough and were thought to be located in the nanopores of the gluten layers [28]. The immobilized water was mainly attributed to the interaction between starch and water molecules [25]. All these results suggested that the interaction force between protein and water and between starch and water was weakened. And the degree of weakening was more pronounced for TFT than for FS. During the thawing process of TFT, due to ice crystal melting and water phase interaction with the dough matrix, the dough’s water-binding capacity was reduced [29]. Moreover, this reduction capacity became more pronounced as the number of cycles of TFT increased. Therefore, the role of TFT on the starch and gluten of dough needs further elaboration.

### 3.5. Pasting Properties

Gelatinization is the basic characteristic of dough. In the gelatinization process of dough, the starch, as the main component of the dough, is decomposed after crushing, which leads to the leaching of amylose, and the starch molecules begin to recrystallize after cooling [30]. In order to better study the effect of the TFT treatment on starch expansion and crystallization destruction in dough, the gelatinization properties of the dough after the TFT treatment were tested.

As shown in Table 1, the pasting temperature (PT) was decreased with the increased number of TFT cycles, indicating that the TFT treatment would make starch in dough easier to gelatinize. Peak viscosity (PV) can reflect the expansion degree and water-retention capacity of starch particles during heating. The result showed that, compared with fresh dough and FS dough, the TFT treatment increased the viscosity of the dough and the value was raised significantly with the increase in cycles. This suggested that TFT may promote the entry of water into the amorphous part of starch particles and further increase the viscosity of the system [21].

The breakdown value can reflect the thermal stability of starch and the leaching degree of amylose during gelatinization. A high breakdown value indicates a low integrity of starch grains [15]. The setback value can reflect the recrystallization degree of gelatinized starch after cooling. Table 1 also shows that the dough under the TFT treatment showed higher breakdown and setback values, and both values were increased with the increase in cycles. This indicated that the thermodynamic stability of starch in the dough was decreased gradually after the TFT treatment. We speculated that the formation and recrystallization of ice crystals during the TFT treatment may cause mechanical damage to starch, leading to the leaching of amylose, thus affecting the crystal structure of starch [17]. In this experiment, the TFT treatment affected the gelatinization properties of starch in dough, destroyed the integrity of starch grains, and promoted the transformation of starch to an ordered structure during cooling.

### 3.6. SDS-PAGE

Gluten aggregation was another character for the dough quality; it exhibited a key role in maintaining water and the physiochemical properties of the starch–gluten network [16]. As shown in Figure 3A, during frozen storage, the strength of the protein bands changed with the different frozen treatments. Compared with the control sample, the brightness of the band in the high-molecular-weight region (72–95 KD) of the sample treated with FS and TFT decreased significantly, but it gradually increased in the low-molecular-weight region (17–55 KD). And the brightness difference was the largest after six times of the TFT treatment. The band changes showed that, with the increased number of TFT cycles, the content of high-molecular-weight protein decreased and the number of low-molecule proteins increased. These phenomena were due to the depolymerization of insoluble high-molecular-weight protein during TFT storage. The depolymerization degree of high-molecular-weight protein was intensified during the TFT treatment, which led to the increase in the number of small-molecule proteins cross-linked by disulfide bonds [19].

Indeed, during the TFT treatment, the gluten network deteriorated because the dough was exposed to extremely fluctuating temperatures. On the one hand, the formation and recrystallization of ice crystals reduced the amount of water bound between protein molecules, and the ice crystals squeezed the protein network structure. These large-molecular-weight protein structures were destroyed and small molecules were aggregated [31]. At the same time, with the increased number of freezing and thawing cycles, the free ice recrystallization and the formation of bubbles in the dough during thawing and freezing caused this trend to be more obvious. On the other hand, the increase in the freezable water content moved the salt ions in the dough to the remaining water, which may lead to the salting-out and denaturation of protein [32]. Based on this, we speculated that protein with higher molecular weight was destroyed under the TFT treatment and it was transformed into protein with a lower molecular weight; this degradation became more obvious with the increased number of freeze–thaw cycles.

It is believed that the changes in the gluten network structure can be attributed to the changes in the secondary structure and protein composition, which involved disulfide bonds, hydrogen bonds, and hydrophobic interactions [33]. Our previous experiments showed that TFT destroyed the rheological properties and protein distribution of dough, which may be associated with the increase in a disordered structure in the protein secondary structure and the destruction of gluten cross-linking. Naturally, this needs further experiments.

### 3.7. SH Content

It was the disulfide bonds that played an important role in maintaining the structural stability of the gluten network. Among them, gluten was a polymer that was formed by intramolecular and intermolecular disulfide bonds [23]. One important indicator of the changes in the disulfide bonds was the change in the free sulfhydryl groups in them.

As shown in Figure 3B, the number of free sulfhydryl groups increased in the dough after both the TFT and FS treatments. Especially after the TFT treatment, the increasement was more drastic. This result was consistent with our SDS-PAGE results; the gluten molecules with a smaller molecular weight could form protein polymers through disulfide bonds, and the change in disulfide bonds was related to the depolymerization of glutenin [31]. The recrystallization of ice crystals under TFT was the main cause of this phenomenon. The TFT treatment led to the breaking of disulfide bonds and directly reduced the contents of the protein polymers. As shown, the free sulfhydryl group increased with the number of cycles under the TFT treatment. This may have been the result of the increasing number of recrystallized ice crystals under the TFT treatment that continuously disrupted the gluten network [34]. Combined with our SDS results, it also showed that the high-molecular-weight gluten proteins were disrupted under TFT, leading to the depolymerization of the glutenin macropolymer owing to the breaking of interchain disulfide bonds during TFT treatment, resulting in more free sulfhydryl groups.

### 3.8. Secondary Structure of Gluten

The changes in the secondary structure can be regarded as markers of protein aggregation, which was mainly maintained through non-covalent binding [35]. In order to further explain the effect of the TFT treatment on protein in frozen dough, the content of the gluten secondary structure after treatment was analyzed from the amide I region.

As shown in Table 2, the quantitative estimation showed that the different protein secondary structure was sensitive to FS and TFT. Both the α-helix and β-turn decreased after freeze treatment, and the contents of the β-sheet and random coil increased. Additionally, the effect of TFT on gluten conformation was significantly higher than that of the FS treatment. The α-helix was the characteristic structure of the alcohol-soluble protein, in which the main maintenance force was the hydrogen bond. The formation and recrystallization of ice crystals in the TFT treatment destroyed the non-covalent hydrogen bond, thus inducing the reduction in the α-helix [17]. When the environment changes, the protein molecules may undergo a conformational rearrangement to achieve the minimum energy to maintain a relatively stable state [36]. As the main structure to maintain the gluten skeleton, the α-helix was relatively orderly. But TFT destroyed the structure and reduced the order of the space. This result was consistent with our dynamic rheology properties.

Additionally, with the increase in the TFT cycles, the β-sheet increased from 26.50% to 28.17%. This indicated that, as ice crystals destroyed the depolymerization and structure of gluten protein during the TFT treatment, the α-helix may transform to the β-sheet with the increase in an unordered structure [37]. In addition, the increase in a random coil further confirmed that the dough was more vulnerable to ice crystal pressure under the TFT treatment. The TFT treatment resulted in the breakage of intra- and intermolecular hydrogen bonds in the starch granules, allowing the partial unfolding and formation of secondary structures [38]. This could affect the aggregation of gluten molecules and significantly increase the relative content of the random coil after the TFT treatment. Therefore, our results showed that the TFT treatment destroyed the molecular structure and inhibited the re-expansion of dough.

### 3.9. SEM

In order to further confirm the effect of TFT on frozen dough, the microstructure of the dough was observed using SEM after different storages (FR, FS, and TFT). As shown in Figure 4, a continuous and smooth structure was observed in the control dough (FR), in which the starch was embedded in the gluten network. However, obvious pores were observed in the dough under FS and TFT treatments, and the internal starch and gluten protein interactions were not as tight as those of the FR. Meanwhile, after the TFT treatment, more starch was exposed in the dough microstructure, especially after six cycles of TFT. This suggested that the FS and TST treatments had a negative effect on the formation of the dough gluten network, and the effect was more obvious in the TFT treatment. In the TFT-treated dough, the gluten network could not support the starch granules well, which led to the splitting and destruction of starch. In the freezing process, ice crystals gradually formed. During the terminal freeze–thaw process, ice crystals regrew and recrystallized. It can be speculated that the TFT treatment would cause more damage to the starch–gluten network and the formation and recrystallization of ice crystals would increase the degree of damage with the increased cycles [39]. Meanwhile, compared to the control group where the starch surface remained relatively smooth, the starch surface was damaged under multiple treatments of TFT and the degree of starch destruction gradually increased with the increased number of cycles. This indicated that the internal and external ice crystals after the TFT treatment exerted some micromechanical force on the starch granules, which enlarged the internal channels of the dough and promoted the leaching of soluble substances [40].

### 3.10. Schematic Model

Based on the results shown above, we proposed a schematic model to elucidate the mechanism of the effects on the moisture, starch, and protein components of frozen dough under the TFT treatment. As shown in Figure 5, after frozen storage, the internal morphology of the dough was significantly different from the FR and this was intensified after the TFT treatment. This was confirmed in our study. In terms of moisture, after the TFT treatment, the amount of freezable water and free water increased and the amount of bound water decreased significantly. These changes in moisture caused mechanical damage and spatial structure changes in the dough. The spatial conformational rearrangement of proteins during frozen storage was caused by the interaction between covalent bonds (S-S) and non-covalent bonds. Our study found that after the TFT treatment, the amount of the covalent disulfide bond decreased significantly, resulting in a significant increase in the free sulfhydryl content. In addition, the formation and recrystallization of ice crystals after the TFT treatment destroyed the structure of the dough, resulting in a decrease in the α-helix and a significant increase in the β-sheet and irregular structure. This eventually led to the destruction of protein polymers with large molecular weights and an increase in the content of proteins with smaller molecular weights. The results of SDS-PAGE also confirmed this conclusion. In terms of starch structure, the ice crystals after the TFT treatment produced mechanical pressure on the starch structure, resulting in the destruction of the starch structure, and, with the increase in the number of cycles, the difference became more obvious. Our gelatinization results and SEM observation further confirmed this conclusion. To sum up, the analysis of the composition of the dough treated with TFT showed that the ice crystals in the dough destroyed the structure of gluten during the TFT cycle and led to the redistribution of water, resulting in the decrease in the gluten cross-linking degree and the destruction of the frozen food structure.

## 4. Conclusions

The effect of TFT on the quality behaviors and component properties of dough were comprehensively shown in this study. Compared with single-frozen storage, we found that the TFT treatment significantly destroyed the texture (increased the hardness and decreased the springiness) and rheological properties (reduced the G′ and G″) of the dough. The differences between only two cycles of TFT and FS treatments were more than the difference between one month of frozen and fresh dough. Additionally, the amount of freezable water and bound water was reduced and the peak viscosity and breakdown value were raised with the increased number of TFT cycles. The protein characteristics showed that the β-sheet and free sulfhydryl groups also showed significantly increased trends during the TFT treatment compared to the single-frozen storage. Our hypothesis was verified: the deterioration of key components in frozen dough under the TFT treatment led to the different dough qualities. These results showed the TFT treatment exhibited a detrimental impact on the dough properties; so, it is essential to control the temperature of the dough after being frozen and stored in a factory in the actual cold chain transportation, especially in hot weather. Our next work is to explore the effect of TFT on baking characteristics and restrain or reduce the quality deterioration under the premise of cost control (such as enzymes, hydrocolloids) so as to increase the yield rate of good products.

## Figures and Tables

**Figure 1 foods-12-03888-f001:**
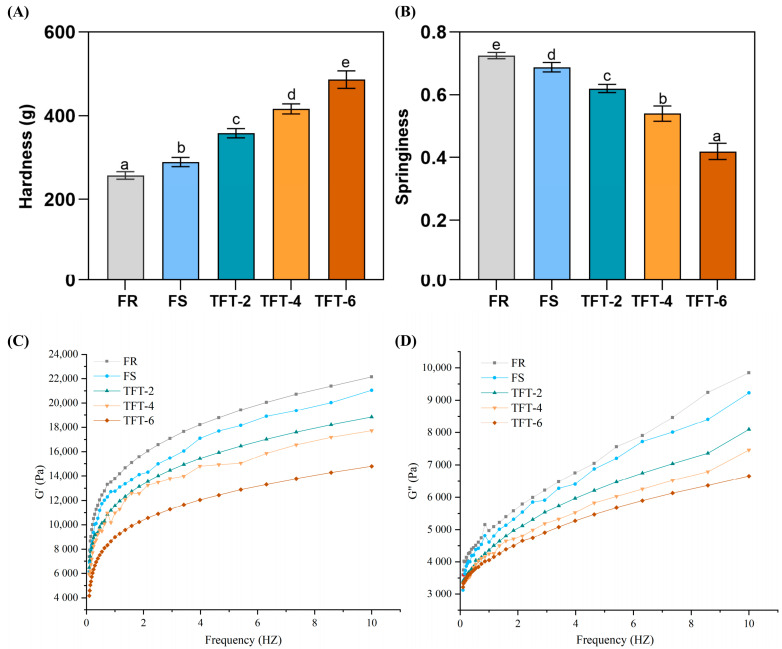
Effect of TFT on the texture and dynamic rheological properties of dough. (**A**,**B**) represent hardness and springiness. (**C**,**D**) represent G′ and G″. Different letters on the bar present significantly different (*p* < 0.05).

**Figure 2 foods-12-03888-f002:**
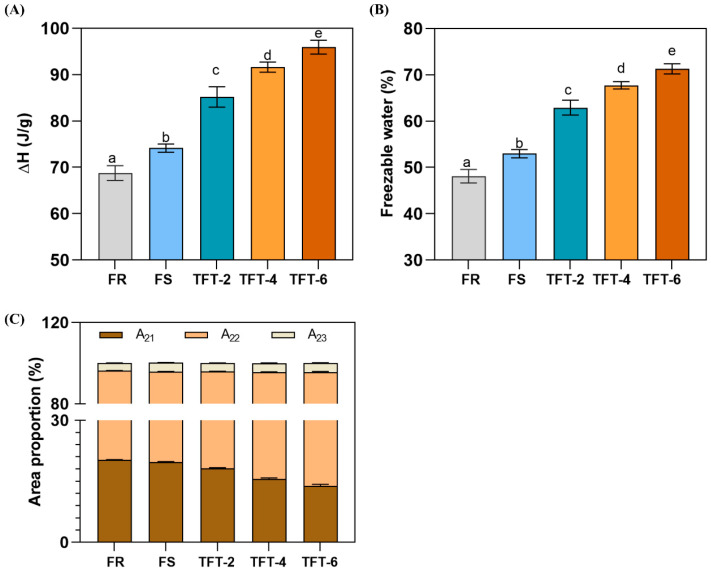
Effect of TFT on the melting enthalpy (**A**), freezable water (**B**), and area proportion (**C**) of dough. Different letters on the bar present significantly different (*p* < 0.05).

**Figure 3 foods-12-03888-f003:**
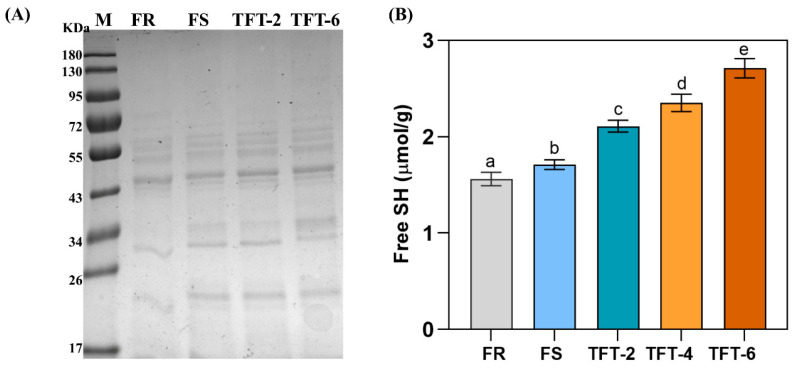
Effect of TFT on the SDS-PAGE analyses (**A**) and free SH (**B**) of dough. Different letters on the bar present significantly different (*p* < 0.05).

**Figure 4 foods-12-03888-f004:**
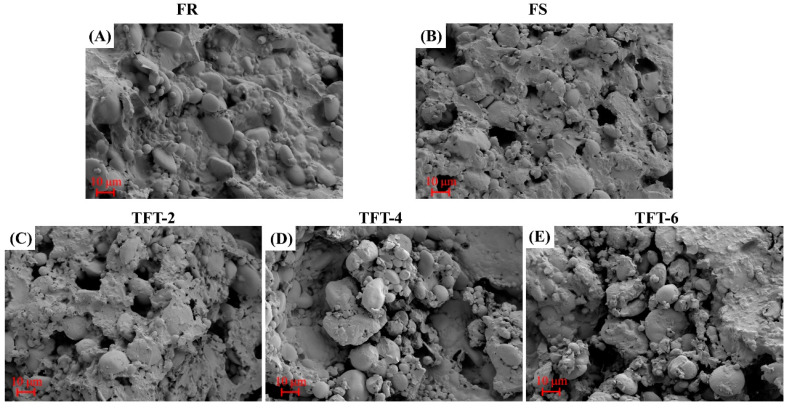
Effect of TFT on the microstructure of dough. (**A**) represents the FR dough. (**B**) represents the dough after FS. (**C**–**E**) represent the dough after TFT treatment for 2, 4, and 6 cycles.

**Figure 5 foods-12-03888-f005:**
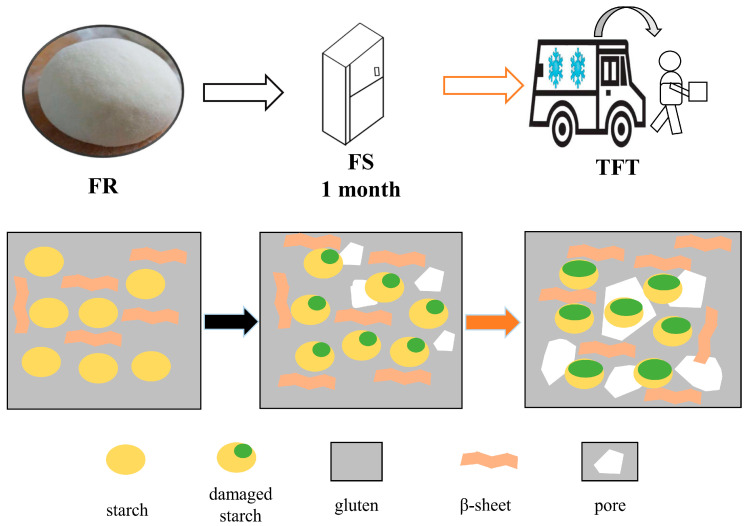
The schematic diagram of dough during TFT treatment.

**Table 1 foods-12-03888-t001:** Effect of TFT treatment on the pasting properties of dough.

Samples	PV (cP)	BD (cP)	ST (cP)	PT (°C)
FR	857 ± 12 ^a^	305 ± 6 ^a^	545 ± 5 ^a^	95.35 ± 0.05 ^d^
FS	919 ± 9 ^b^	320 ± 7 ^b^	574 ± 4 ^b^	95.25 ± 0.06 ^d^
TFT-2	960 ± 15 ^c^	332 ± 8 ^c^	644 ± 8 ^c^	95.00 ± 0.02 ^c^
TFT-4	991 ± 11 ^d^	394 ± 6 ^d^	665 ± 9 ^d^	94.50 ± 0.05 ^b^
TFT-6	1134 ± 12 ^e^	423 ± 8 ^e^	718 ± 7 ^e^	93.50 ± 0.04 ^a^

PV: Peak viscosity. BD: Breakdown viscosity. ST: Setback viscosity. PT: Pasting temperature. Data are expressed as means ± SD of duplicate assays. Values followed by different superscripts in the same column present significantly different (*p* < 0.05).

**Table 2 foods-12-03888-t002:** Effect of TFT treatment on the secondary structure contents of dough.

Samples	Secondary Structure (%)			
	α-Helix	β-Sheet	β-Turn	Random Coil
FR	17.53 ± 0.06 ^d^	26.50 ± 0.10 ^a^	40.37 ± 0.21 ^d^	15.60 ± 0.12 ^a^
FS	17.42 ± 0.09 ^d^	26.87 ± 0.13 ^b^	39.96 ± 0.12 ^c^	15.75 ± 0.11 ^a^
TFT-2	17.37 ± 0.08 ^c^	27.27 ± 0.09 ^c^	39.02 ± 0.13 ^b^	16.34 ± 0.13 ^b^
TFT-4	17.16 ± 0.13 ^b^	27.79 ± 0.18 ^d^	38.57 ± 0.19 ^a^	16.48 ± 0.09 ^b^
TFT-6	16.95 ± 0.08 ^a^	28.17 ± 0.13 ^e^	38.23 ± 0.15 ^a^	16.65 ± 0.10 ^c^

Data are expressed as means ± SD of duplicate assays. Values followed by different superscripts in the same column present significantly different (*p* < 0.05).

## Data Availability

The data presented in this study are available on request from the corresponding author.

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
