# Peer review of "The Effect of Terminal Freezing and Thawing on the Quality of Frozen Dough: From the View of Water, Starch, and Protein Properties"

_foods, 2023, doi:10.3390/foods12213888_

Round 1

Reviewer 1 Report

The presented material seems interesting and requires partial improvement.

The authors need to make minor adjustments to improve the article.

The comments are presented below.

1. Present the scientific purpose of the research in more detail and specificity in the abstract of the article

2. Present a generalized research scheme

3. It is necessary to list the equipment (describe the brand and country of manufacturer) used to obtain the test in paragraph 2.2)

4. Create a more complete description of samples (TFT-2, TFT-4, TFT-6)

5. Clause 3.4. “Textural properties of the dough...” not sufficiently detailed.

6. In conclusion, it is necessary to indicate possible options for the practical application of the results obtained, taking into account the financial costs of the proposed technology

Author Response

The presented material seems interesting and requires partial improvement. The authors need to make minor adjustments to improve the article. The comments are presented below.

Reply: We feel great thanks for your helpful and constructive comments on our manuscript. As you are concerned, there are several concerns that need to be addressed. According to your nice suggestions, we have made extensive corrections to our previous draft.

All revised portions are marked in red in the manuscript and the detailed responses are as follows.

Question 1-1: Present the scientific purpose of the research in more detail and specificity in the abstract of the article

Reply: Thanks for your good suggestion. We have modified it in our revised manuscript (Please see Abstract).

Question 1-2: Present a generalized research scheme

Reply: Thank you for pointing out this. We have modified it in our revised manuscript (Please see lines 16-19, 76-81).

Question 1-3: It is necessary to list the equipment (describe the brand and country of manufacturer) used to obtain the test in paragraph 2.2)

Reply: We sincerely thank the reviewer for careful reading. As suggested by the reviewer, we have added the equipment information (Please see paragraph 2.2).

Question 1-4: Create a more complete description of samples (TFT-2, TFT-4, TFT-6)

Reply: Thank you for pointing out this. We have added the description (Please see lines 99-101).

Question 1-5: Clause 3.4. “Textural properties of the dough...” not sufficiently detailed.

Reply: We sincerely thank the reviewer for careful reading and have modified it (Please see line 245).

Question 1-6: In conclusion, it is necessary to indicate possible options for the practical application of the results obtained, taking into account the financial costs of the proposed technology

Reply: Thank you for your comment. We have added some description about the application and our next work (Please see lines 458-461).

Once again, special thanks to you for your good comments, which really help us to improve the manuscript. Based on your suggestions, we tried our best to improve the manuscript and made some changes in the manuscript. We hope that the revised manuscript will be fine.

Reviewer 2 Report

The manuscript of Xiaorong Liu  et al reports interesting data. This article has high research value. The use of frozen dough on a mass scale has become very popular in recent years. The subject matter is very up-to-date. Concerning methodology, the methods used for the experimental part of the study are up to date and relevant. The results are well-discussed. However, this article requires some minorcorrections. The comments are included below.

 - Abstract: ‘ In conclusion, the research was helpful to provide guidance for the cold chain transportation industry of frozen dough.’ - Please specify what guidelines have been established

- The purpose of the work is clearly formulated.

- The obtained results were satisfactorily discussed with the available literature.

- Conclusions correct.

- From a practical point of view, it is also proposed to perform laboratory baking in future research. Complementing research in this area would be of significant cognitive and practical importance.

Author Response

The manuscript of Xiaorong Liu et al reports interesting data. This article has high research value. The use of frozen dough on a mass scale has become very popular in recent years. The subject matter is very up-to-date. Concerning methodology, the methods used for the experimental part of the study are up to date and relevant. The results are well-discussed. However, this article requires some minor corrections. The comments are included below.

Reply: Thank you very much for your attention and evaluation on our paper. We have revised the manuscript according to your kind advices and suggestions to improve the quality of our manuscript. All revised portions are marked in red in the manuscript and the detailed responses are as follows.

Question 2-1: Abstract: ‘ In conclusion, the research was helpful to provide guidance for the cold chain transportation industry of frozen dough.’ - Please specify what guidelines have been established

Reply: Thank you for your valuable comments and suggestions. We have rewritten this sentence (Please see lines 26-28).

Question 2-2: The purpose of the work is clearly formulated.

Reply: Thank you for your comments, which give us great encouragement.

Question 2-3: The obtained results were satisfactorily discussed with the available literature.

Reply: Thanks for your careful checks.

Question 2-4: Conclusions correct.

Reply: We sincerely thank the reviewer for careful reading.

Question 2-5: From a practical point of view, it is also proposed to perform laboratory baking in future research. Complementing research in this area would be of significant cognitive and practical importance.

Reply: Thank you for your comment. We have added some description about the application and our next work (Please see lines 455-461).

Once again, special thanks to you for your good comments, which really help us to improve the manuscript. Based on your suggestions, we tried our best to improve the manuscript and made some changes in the manuscript. We appreciate for your warm work earnestly, and hope that the correction will meet with approval.

Reviewer 3 Report

First of all, keywords are not well-defined as they are the same as the title. They should increase the recognisability of the article among thousands of others and increase its positioning. Right now their amount is limited, long, and does not precisely define the topic.

The authors said, "In conclusion, the research was helpful to provide guidance for the cold chain transportation industry of frozen dough." Instead of "to provide" to "in providing". So what are the instructions and advice to keep the dough in good condition? That should be highlighted in the conclusion part. 

I expect that not only the type of freezing will be changed but also the usage of different yeasts will be proposed. 

What is the innovation in this technique of freezing? Is it used in the industry? in my opinion, it should give more insights into innovative solutions e.g. energy saving, time and duration of possible transportation in the not destructive matter, type of packaging, etc. the manuscript contains a wide range of standard methods to determine the physical properties of dough, however, doesn't have a good conclusion.

A lot of editor corrections should be done e.g. upper cases in the sign of degrees Celsius or "cm-1" etc.

Line 142: name and producer of the instrument that was used to measure "Free thiol group (SH) content".

I am not qualified to judge the English grammar.

Author Response

Question 3-1: First of all, keywords are not well-defined as they are the same as the title. They should increase the recognisability of the article among thousands of others and increase its positioning. Right now their amount is limited, long, and does not precisely define the topic.

Reply: We sincerely thank the reviewer for careful reading. We have revised the keywords based on your suggestion (Please see Keywords).

Question 3-2: The authors said, "In conclusion, the research was helpful to provide guidance for the cold chain transportation industry of frozen dough." Instead of "to provide" to "in providing". So what are the instructions and advice to keep the dough in good condition? That should be highlighted in the conclusion part.

Reply: Thank you for pointing out this. We have added some description about instructions and advice to keep the dough in good condition (Please see the conclusion part).

Question 3-3: I expect that not only the type of freezing will be changed but also the usage of different yeasts will be proposed.

Reply: We greatly appreciate for your comment. At present, this work was focused on the different freezing treatment types, from different aspect to expatiate the change difference. Additionally, we also agree with your comment. So in the next work, we will investigate the changes of frozen dough with different ingredients under different freezing types, such as different yeasts.

Question 3-4: What is the innovation in this technique of freezing? Is it used in the industry? in my opinion, it should give more insights into innovative solutions e.g. energy saving, time and duration of possible transportation in the not destructive matter, type of packaging, etc. the manuscript contains a wide range of standard methods to determine the physical properties of dough, however, doesn't have a good conclusion.

Reply: Thank you for your valuable comment. When the dough is produced in the central factory, after a period of frozen storage, it will be transported to the store by transport trucks. When the weather is hot, during this terminal transportation or transfer process, due to the control of mechanical or equipment costs, the dough often faces sharp fluctuations in temperature. For example, in Wuhan, where the author lives, the summer temperature is close to 40℃, which will lead to a serious reduction in the yield of frozen dough. This topic is also extracted from the actual process of the factory, and then we would restrain or reduce the quality deterioration during TFT in our next work. We sincerely thank the reviewer and have rewritten the conclusions based on your suggestion (Please see the conclusion part).

Question 3-5: A lot of editor corrections should be done e.g. upper cases in the sign of degrees Celsius or "cm-1" etc.

Reply: We were sorry for the careless mistake and we have corrected it (Please see lines 93-98, 114-115, 129-131, 154).

Question 3-6: Line 142: name and producer of the instrument that was used to measure "Free thiol group (SH) content".

Reply: Thank you for pointing out this. We have added the name and producer of the instrument (Please see lines 149-150).

Once again, special thanks to you for your good comments, which really help us to improve the manuscript. Based on your suggestions, we tried our best to improve the manuscript and made some changes in the manuscript. We appreciate for your warm work earnestly, and hope that the correction will meet with approval.

Reviewer 4 Report

This manuscript reports about the freezing/thawing behavior of dough. However, this topic is not novel. So, novelty needs to be explained or highlighted much more in detail. In recent years, so many articles were already published in that field, describing all kinds of doughs/products such as noodels, bread, sumplings etc.. There need to be a clearer differentiation from what has done before. A clear Scientific hypothesis is missing. A statement such as "Based on this background, the study on the physical and chemical properties of frozen dough under the terminal freezing and thawing" is too general...

A certain revision of language is also needed. E.g., "....this paper aims to investigate...." is not correct. Papers cannot do anything, reserachers do....

Please check formats....description on temepreature values are wrong. It is not "oC"...

What is "trigger force was 5 gf" ?

Figures are by far too small. They cannot be read properly....labelling is hardly visible. (Here, I often I do not understand what authors think of when submitting things like these....)

Figure 5 is not really self-explanatorily.

Conclusion is more like a summary of a review article. All these facts are almost already known from the literature, even some of them from textbook knowledge. Novelty ? More (novel) feasible conclusions needed.

Language needs significant improvement.

Author Response

Question 4-1: This manuscript reports about the freezing/thawing behavior of dough. However, this topic is not novel. So, novelty needs to be explained or highlighted much more in detail. In recent years, so many articles were already published in that field, describing all kinds of doughs/products such as noodles, bread, dumplings etc.. There need to be a clearer differentiation from what has done before. A clear Scientific hypothesis is missing. A statement such as "Based on this background, the study on the physical and chemical properties of frozen dough under the terminal freezing and thawing" is too general...

Reply: Thank you very much for your attention and patience on our paper. We agree with you that many studies in recent years have focused on freezing or freeze-thaw cycles, but it is more focused on the single treatment. However, in actual production, in many cases, the freezing or freeze-thaw cycle does not exist independently. When the dough is produced in the central factory, after a period of frozen storage, it will be transported to the store by transport trucks. When the weather is hot, during this terminal transportation or transfer process, due to the control of mechanical or equipment costs, the dough often faces sharp fluctuations in temperature. For example, in Wuhan, where the author lives, the summer temperature is close to 40℃, which would lead to a serious reduction in the yield of frozen dough.

       Our study focuses on the differences in mass changes after this composite treatment, rather than individual freezing or thawing. This kind of treatment is more complex with the actual factory situation, and this is also the problem we found when we went to the enterprise to investigate. That is, the dough goes through the same pretreatment and frozen storage, but the yield rate in summer was significantly lower. Therefore, we focus on the terminal temperature change after frozen storage, and compares the difference between this process and the single frozen storage.

Meanwhile, we sincerely thank the reviewer for careful reading. We have modified the novelty and added the hypothesis in our revised manuscript (Please see lines 68-81).

Question 4-2: A certain revision of language is also needed. E.g., "....this paper aims to investigate...." is not correct. Papers cannot do anything, researchers do....

Reply: We were sorry for the careless mistake and we have corrected it. Meanwhile, we have modified the language in our revised manuscript.

Question 4-3: Please check formats....description on temperature values are wrong. It is not "oC"...

Reply: We were sorry for the careless mistake and we have corrected it (Please see lines 93-98, 114-115, 129-131, 154).

Question 4-4: What is "trigger force was 5 gf" ?

Reply: We gratefully appreciate for your comment. When the probe of the texture analyzer is compressed down and slowly approaches the sample, and the force between the probe and the sample reaches the set trigger force, it is considered that the probe has contacted the sample, and the experiment can be started. To make it more clearly, we have modified this description (Please see lines 104-105).

Question 4-5: Figures are by far too small. They cannot be read properly....labelling is hardly visible. (Here, I often I do not understand what authors think of when submitting things like these....)

Reply: We are sorry for this. We have revised it according to your comment (Please see Figures).

Question 4-6: Figure 5 is not really self-explanatorily.

Reply: Thank you for pointing out this. We have modified the Figure 5 to make it more clearly.

Question 4-7: Conclusion is more like a summary of a review article. All these facts are almost already known from the literature, even some of them from textbook knowledge. Novelty ? More (novel) feasible conclusions needed.

Reply: Thanks for your comments, we have revised the conclusion to show our highlight and novelty as much as possible (Please see lines 445-461).

Once again, special thanks to you for your good comments, which really help us to improve the manuscript. Based on your suggestions, we tried our best to improve the manuscript and made some changes in the manuscript. We appreciate for your warm work earnestly, and hope that the correction will meet with approval.

Round 2

Reviewer 3 Report

The Authors took into consideration my last comments, however, I have one more.

If possible some numerical findings should be added to the abstract part. e.g. Line 19 "significantly increased", Line 20: "decreased" and "increased", and Line 21: "TFT increased", but how much?

Author Response

The Authors took into consideration my last comments, however, I have one more.

If possible some numerical findings should be added to the abstract part. e.g. Line 19 "significantly increased", Line 20: "decreased" and "increased", and Line 21: "TFT increased", but how much?

Reply: We sincerely thank the reviewer for careful reading. Based on your suggestion, we have added numerical findings on the basis of controlling the total number of abstract words (Please see Abstract).

We appreciate for your warm work earnestly, and hope that the correction will meet with approval.

Reviewer 4 Report

Revision lead to a certain improvement of the manuscript. However, Especially the revised parts brought some new mistakes in language. This needs still improvement.

Especially the revised parts brought some new mistakes in language. This needs still improvement.

Author Response

Revision lead to a certain improvement of the manuscript. However, Especially the revised parts brought some new mistakes in language. This needs still improvement.

Reply: Thank you very much for your attention and patience on our paper. We have revised several mistakes in language.

We appreciate for your warm work earnestly, and hope that the correction will meet with approval.